# MathCAMPS: Fine-grained Synthesis of Mathematical Problems From Human Curricula

**Shubhra Mishra**[*,1]   **Gabriel Poesia**[*,1]   **Belinda Mo**[1]   **Noah D. Goodman**[1,2]

{shubhra,poesia,ngoodman}@stanford.edu     bmo98@alumni.stanford.edu

Departments of Computer Science[1] and Psychology[2], Stanford University

## Abstract

Mathematical problem solving is an important skill for Large Language Models (LLMs), both as an important capability and a proxy for a range of reasoning abilities. Existing benchmarks probe a diverse set of skills, but they yield aggregate accuracy metrics, obscuring specific abilities or weaknesses. Furthermore, they are difficult to extend with new problems, risking data contamination over time. To address these challenges, we propose MathCAMPS: a method to synthesize high-quality mathematical problems at scale, grounded on 44 fine-grained "standards" from the Mathematics Common Core (CC) Standard for K-8 grades. We encode each standard in a formal grammar, allowing us to sample diverse symbolic problems and their answers. We then use LLMs to realize the symbolic problems into word problems. We propose a cycle-consistency method for validating problem faithfulness. Finally, we derive *follow-up questions* from symbolic structures and convert them into follow-up word problems—a novel task of mathematical dialogue that probes for robustness in understanding. Experiments on 23 LLMs show surprising failures even in the strongest models (in particular when asked simple follow-up questions). Moreover, we evaluate training checkpoints of Pythia 12B on MathCAMPS, allowing us to analyze when particular mathematical skills develop during its training. Our framework enables the community to reproduce and extend our pipeline for a fraction of the typical cost of building new high-quality datasets. Project page: https://mathcamps.cc.

## 1  Introduction

As Large Language Models (LLMs) become increasingly capable, mathematical reasoning has become a key benchmark for evaluating their abilities. Traditional benchmarking, which relies on fixed sets of human-generated problems (e.g., GSM8k[8], or MATH [11]), now faces new challenges. Many LLMs are trained on vast public datasets that may include these benchmarks, raising concerns about data contamination [20, 7, 4]. This issue is amplified by the lack of transparency in the training data of most state-of-the-art models, including GPT-4 [1], Claude [2], and LLaMA [19]. While creating novel problems could mitigate contamination concerns but is resource-intensive. Moreover, current benchmarks offer limited insights into the specific mathematical skills of LLMs, as aggregate accuracy alone does not reveal where models excel or struggle, and how this has changed over time.

To address these issues, we introduce the Mathematics Common Core Assessment of Problem Solving (MathCAMPS), a framework for generating high-quality mathematical problems based on the Common Core (CC) standards. MathCAMPS enables detailed analysis of LLMs' mathematical proficiency, aligned with skills taught in schools. Our pipeline employs a composable grammar for generating problems, symbolic solvers (e.g. SymPy) to get final solutions, and an LLM for transforming them into word problems. We ensure problem faithfulness through a cycle-consistency check, where the LLM back-translates word problems into symbolic form.

38th Conference on Neural Information Processing Systems (NeurIPS 2024).

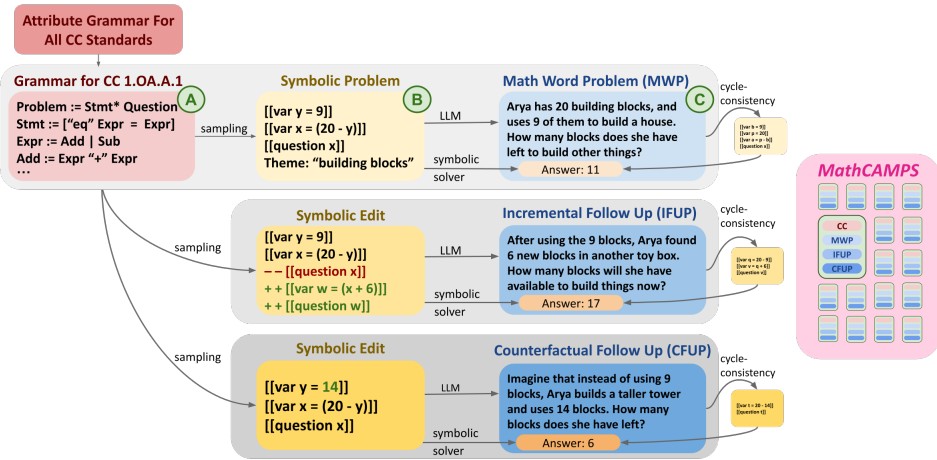

Figure 1: Overview of the MathCAMPS generation pipeline. We start from a grammar (**A**) that represents problems tied to a Common Core Standard - a specific mathematical ability drawn from a human curriculum. We sample problems in a symbolic form (**B**), and use a language model to realize it in natural language (**C**), applying a cycle-consistency where we back-translate the problem into symbolic form and ensure the answer remains the same, validating truthfulness. We also synthesize incremental and counterfactual follow-up problems

We also propose a novel "mathematical dialogue" task, where the model answers follow-up questions after solving a problem. These follow-ups can be either *counterfactual*, modifying an aspect of the original problem, or *incremental*, providing additional information and changing the question.

Using our framework, we synthesize problems for each of 44 CC standards (Appendix C), resulting in a dataset of 4,900 initial problems and 4707 total follow-ups. Our results reveal surprising weaknesses, particularly in response to follow-up responses, highlighting significant gaps in even the strongest models. Additionally, we provide a first-of-its-kind analysis of learning dynamics of mathematical abilities in LLM training using checkpoints from Pythia 12B [6] (Appendix B).

## 2 MathCAMPS

We now describe our pipeline for automatically generating mathematical problems and follow-up questions that are grounded in a human curriculum – the Mathematics Common Core (`https://www.thecorestandards.org`). Figure 1 overviews our pipeline. We describe how we represent CC standards in a grammar, sample symbolic problems, generate follow-ups, realize those in natural language, and finally improve quality by checking for cycle consistency.

**Representing Common Core Standards** We represent CC standards using an attribute grammar [10], allowing both syntactic and semantic rules. This formalism supports context-sensitive constraints, enabling encoding of information like numerical bounds directly in production rules.

**From Symbolic to Word Problems** To convert symbolic problems into natural language, we use few-shot prompting with GPT-4 (Figure 1 (C)). For each standard, we manually create word problems from two symbolic examples. For word problems requiring cover stories, we randomly select a theme from a set of 188. These examples guide GPT-4 in generating diverse, natural problems. To ensure faithfulness to the original structure, we apply a *cycle consistency* approach: GPT-4 converts its generated word problem back into a symbolic structure, which is solved and compared to the original. Problems failing this test are discarded.

**Generating Follow-Up Questions** We leverage symbolic representations to generate two types of follow-up questions: *counterfactual* (altering a constant) and *incremental* (adding information). For each CC standard, we identify applicable follow-up types. Symbolically, follow-up questions are modeled as differences applied to the original problem, which we solve to produce ground-truth answers. We use few-shot prompting to translate these changes into natural language questions and apply cycle consistency to verify accuracy.

Table 1: Final answer accuracy of LLMs on MathCAMPS, both over all problems (**All**) and considering only standards in each grade we cover (**K** to **8**). Highlights compare to gradewise avg.

| Vendor | Model | All | K | 1 | 2 | 3 | 4 | 5 | 6 | 7 | 8 |
|--------|-------|-----|---|---|---|---|---|---|---|---|---|
| OpenAI | GPT-4o [1] | 0.92 | 0.98 | 0.98 | 0.98 | 0.98 | 0.92 | 0.88 | 0.95 | 0.89 | 0.64 |
| Anthropic | Claude-3 Opus [2] | 0.89 | 0.97 | 0.99 | 0.96 | 0.98 | 0.89 | 0.83 | 0.96 | 0.73 | 0.56 |
| Google | Gemini-1.5 Pro [17] | 0.89 | 0.95 | 0.98 | 0.97 | 0.97 | 0.89 | 0.83 | 0.93 | 0.78 | 0.54 |
| Google | Gemini-1.5 Flash [17] | 0.87 | 0.98 | 0.98 | 0.97 | 0.98 | 0.80 | 0.80 | 0.90 | 0.84 | 0.56 |
| OpenAI | GPT-3.5 Turbo [1] | 0.87 | 0.96 | 0.98 | 0.98 | 0.97 | 0.86 | 0.77 | 0.90 | 0.77 | 0.56 |
| Anthropic | Claude-3 Sonnet [2] | 0.86 | 0.96 | 0.98 | 0.97 | 0.98 | 0.88 | 0.74 | 0.94 | 0.66 | 0.49 |
| Anthropic | Claude-3 Haiku [2] | 0.84 | 0.97 | 0.98 | 0.97 | 0.98 | 0.87 | 0.69 | 0.92 | 0.59 | 0.51 |
| Meta | Llama 3 70B [19] | 0.85 | 0.96 | 0.97 | 0.97 | 0.97 | 0.85 | 0.71 | 0.87 | 0.73 | 0.50 |
| Mistral | Mixtral 8x22B [13] | 0.84 | 0.96 | 0.99 | 0.98 | 0.96 | 0.79 | 0.69 | 0.88 | 0.73 | 0.61 |
| DeepSeek | DeepSeek 67B [5] | 0.80 | 0.95 | 0.99 | 0.96 | 0.93 | 0.82 | 0.60 | 0.84 | 0.61 | 0.47 |
| Meta | Llama 3 8B [19] | 0.77 | 0.94 | 0.97 | 0.96 | 0.94 | 0.78 | 0.55 | 0.79 | 0.53 | 0.43 |
| Mistral | Mixtral 8x7B [13] | 0.76 | 0.94 | 0.96 | 0.93 | 0.91 | 0.75 | 0.52 | 0.80 | 0.53 | 0.45 |
| EleutherAI | Llemma 34B [3] | 0.71 | 0.95 | 0.96 | 0.93 | 0.87 | 0.61 | 0.47 | 0.77 | 0.46 | 0.44 |
| Mistral | Mistral 7B [12] | 0.68 | 0.89 | 0.94 | 0.91 | 0.84 | 0.61 | 0.42 | 0.66 | 0.45 | 0.42 |
| DeepSeek | DeepSeek Coder 33B [9] | 0.65 | 0.88 | 0.93 | 0.92 | 0.83 | 0.54 | 0.36 | 0.66 | 0.44 | 0.38 |
| Meta | CodeLlama 34B [15] | 0.64 | 0.90 | 0.94 | 0.92 | 0.85 | 0.51 | 0.38 | 0.70 | 0.37 | 0.30 |
| Microsoft | phi-2 [14] | 0.63 | 0.95 | 0.96 | 0.89 | 0.78 | 0.46 | 0.38 | 0.61 | 0.37 | 0.41 |
| EleutherAI | Llemma 7B [3] | 0.62 | 0.88 | 0.90 | 0.85 | 0.79 | 0.48 | 0.41 | 0.67 | 0.41 | 0.36 |
| Google | Gemma 7B [18] | 0.62 | 0.83 | 0.92 | 0.90 | 0.82 | 0.47 | 0.36 | 0.65 | 0.36 | 0.30 |
| Meta | CodeLlama 13B [15] | 0.58 | 0.87 | 0.92 | 0.87 | 0.75 | 0.41 | 0.30 | 0.61 | 0.32 | 0.34 |
| Meta | CodeLlama 7B [15] | 0.52 | 0.85 | 0.92 | 0.84 | 0.69 | 0.37 | 0.25 | 0.57 | 0.25 | 0.16 |
| Google | Gemma 2B [18] | 0.51 | 0.66 | 0.76 | 0.74 | 0.67 | 0.42 | 0.28 | 0.55 | 0.30 | 0.27 |
| - | Avg. Performance | 0.74 | 0.87 | 0.91 | 0.89 | 0.87 | 0.70 | 0.59 | 0.78 | 0.57 | 0.38 |

## 3 Experiments

We now evaluate a suite of 23 LLMs from 8 different vendors on MathCAMPS. We evaluate all models by sampling with temperature 0, using a fixed 1-shot prompt with the first example from GSM8K, mostly to demonstrate the format. For all models (most of them instruction-tuned), a single example was enough for to adhere to the task and the format we specify. The rich structure in MathCAMPS allows us to perform a number of unique analyses on LLMs relating to specific mathematical abilities and their corresponding grade levels for human students.

Table 1 shows both aggregate accuracy on MathCAMPS, as well as accuracy across standards partitioned by grade, whereas Figure 3 compares the aggregate accuracies on MathCAMPS and GSM8K. Closed-weights models are shown above the line, with open-weights models below. GPT-4o ranks at the top in overall accuracy. Since we used GPT-4 to generate the problems, we must rule out familiarity bias [16] in this result, which we do in Appendix D.

**Models of similar overall performance can have large disparities in specific abilities or grades.** Several models that have comparable overall accuracies show large differences when compared on specific mathematical skills. As an example, Claude-3 Opus and Claude-3 Sonnet have similar overall accuracy both in MathCAMPS (.89 vs .86) and in GSM8K (.95 vs .923). However, we find that Claude-3 Opus is significantly better at manipulating fractions. For instance, in the CC standard 5.NF.A.2, described as *"Solve word problems involving addition and subtraction of fractions referring to the same whole, including cases of unlike denominators"*, Opus has a 36% advantage over Sonnet, scoring a 70% accuracy for this standard, whereas Sonnet only achieves 34%. Similarly, while Gemma 7B and phi-2 have comparable overall performance (.62 vs .63 accuracy on MathCAMPS), some capabilities in each model seem nearly absent from the other. Gemma 7B is highly accurate when performing multi-digit multiplication (4.NBT.B.4), which phi-2 struggles with. And while phi-2 performs well while comparing fractions (4.NF.A.2), Gemma 7B struggles. Such stark differences are obscured when only analyzing aggregate metrics, whereas MathCAMPS allows for a much more nuanced understanding of mathematical reasoning capabilities.

**Overall ranking between models is largely a function of which skills we choose to evaluate.** Overall accuracies in any dataset induce a single performance ranking of models. However, when we look at individual CC standards in MathCAMPS, rankings are largely a function of which skills we choose to evaluate. Comparing pairs of models across all standards, rarely we find cases where

one model Pareto-dominates another (i.e. is better on all standards): only 23.08% of all pairs of models have a Pareto winner. Table 3 shows how the ranking of a model in individual skills can often deviate strongly from its overall ranking. Here, the first ordinal in each cell shows the model's ranking on overall performance on MathCAMPS, whereas the second shows the model's ranking on that particular CC standard. We find many cases of large discrepancies. For instance, on systems of equations, GPT-4o tends to excessively rely on decimal approximations when operating with fractions, resulting in poor performance. Llemma 34B, which places 13th overall, is the best performing model on a simple kindergarten-level word problems on adding to complete 10.

**Follow-up tasks** We now evaluate the performance of LLMs on follow-up questions. Here, we first give a problem, and in case the model answers correctly we ask either an incremental follow-up, a counterfactual follow-up, or both (in separate contexts), depending on the standard (some standards don't have follow-ups, and for some problems we failed to find a cycle-consistent follow-up within the max attempts). Here, we're interested in analyzing the (lack of) robustness that LMs might have when probed with extra questions — our follow-ups are generally answerable using the same core mathematical knowledge involved in the initial problem but require longer range attention and dialog understanding.

Table 3 (full table with all models in the Appendix) shows overall accuracies when we only consider a model successful on a problem when it also answers its follow-up questions correctly. We also show the major accuracy drops across CC standards for each model (last two columns). We find many notable cases, in both stronger and weaker models. GPT-4o, for instance, is 90% accurate in evaluating expressions of addition of fractions with multi-digit numerators and denominators (`5.NF.A.1` — notably, this requires putting fractions in the same denominator). When asked to add another fraction to the result, or change one of the original fractions to a new one and re-do the computation, its success rate when evaluated at correctly answering both follow-ups drops to 61%, or a 29% decrease. Other models drop even more dramatically. For instance, phi-2 solves 57% of the problems in `7.NS.A.2`, which are about multiplying two fractions (only requires two multi-digit multiplications — we do not require the result to be in lowest terms). However, when asked to multiply the result by a further third fraction, phi-2 tends to not reuse its previous (correct) result, and instead proceeds by writing down the product of the three numerators (and denominators), and attempt to directly evaluate this product. This strategy is rarely successful, and it only achieves 8% accuracy when accounting for the follow-ups (an absolute 49% drop). Overall, we find many cases where models are not robust to simple follow-up questions. We hypothesize that this setup of mathematical dialogue is much less frequent in pre-training data, and that follow-up problems in MathCAMPS can be a rich source of further analyses for future work.

Table 2: Model performance on our mathematical dialogue task, where the model must answer follow-up questions besides the initial problem. Results for all models are shown in the Appendix.

| Model | Acc. with follow-ups | Largest accuracy drop w/ follow-ups | |
| --- | --- | --- | --- |
| GPT-4o | 0.82 | 5.NF.A.1 - Add/sub fractions | 0.90 ↘0.61) |
| Claude-3 Opus | 0.76 | 7.NS.A.1-fraction - Add/sub with fractions | 0.57 ↘0.25) |
| Gemini-1.5 Pro | 0.77 | 5.NF.A.1 - Add/sub fractions | 0.60 ↘0.35) |
| GPT-3.5 Turbo | 0.71 | 7.NS.A.1-fraction - Add/sub with fractions | 0.73 ↘0.22) |
| Llama 3 70B | 0.69 | 4.NF.A.2 - Compare two fractions | 0.99 ↘0.66) |
| Mixtral 8x22B | 0.69 | 7.NS.A.1-fraction - Add/sub with fractions | 0.69 ↘0.18) |
| DeepSeek 67B | 0.68 | 6.NS.B.3 - Add/sub/mult/div decimals | 0.59 ↘0.37) |
| phi-2 | 0.39 | 7.NS.A.2 - Mult/div with fractions | 0.57 ↘0.08) |
| Gemma 7B | 0.33 | 7.NS.A.1-decimal - Add/sub with decimals | 0.91 ↘0.32) |

## 4   Conclusion

We introduce MathCAMPS, a fine-grained synthetic benchmark of mathematical reasoning in LLMs. MathCAMPS is directly grounded on the Common Core Standards, a widely used curriculum in human education. By tying our problems to a human curriculum, we enable a much wider range of analyses to understand mathematical reasoning capabilities and weaknesses of LLMs. We show analyses of performance by grade level and identify particularly challenging skills for a range of models, though we believe these are only a few examples of analyses that MathCAMPS permits.

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

# A Tables

Table 3: Largest model rank changes when focusing on one CC standard. Here, A ↗B indicates that the model ranks $A^{th}$ on MathCAMPS overall, but ranks $B^{th}$ when only evaluating on problems from the indicated CC standard. Conversely, ↘marks notable cases where a model's performance on the indicated CC standard is lower than its overall performance on MathCAMPS. We show selected rows here, the complete table can be found in the Appendix.

| Model | Top outlier skill | Rank change |
|---|---|---|
| GPT-4o | 8.EE.C.8 - Solve two-variable systems | $(1^{st}$ ↘$22^{th})$ |
| Claude-3 Opus | 2.MD.B.5 - Add/sub within 100 | $(2^{nd}$ ↘$13^{th})$ |
| Gemini-1.5 Pro | K.OA.A.4 - Adding to equal 10 | $(3^{rd}$ ↘$19^{th})$ |
| Gemini-1.5 Flash | 4.OA.B.4 - Factor pairs within 100 | $(4^{th}$ ↘$20^{th})$ |
| Claude-3 Haiku | 3.OA.A.4 - Determine unknowns in mul/div probs | $(9^{th}$ ↗$1^{st})$ |
| Llama 3 70B | K.OA.A.4 - Adding to equal 10 | $(7^{th}$ ↘$17^{th})$ |
| DeepSeek 67B | K.NBT.A.1 - Decompose into 10s | $(10^{th}$ ↗$1^{st})$ |
| Llemma 34B | K.OA.A.4 - Adding to equal 10 | $(13^{th}$ ↗$1^{st})$ |
| Mistral 7B | 1.OA.A.1 - Add/sub within 20 | $(14^{th}$ ↘$21^{th})$ |
| DeepSeek Coder 33B | 6.EE.A.1 - Evaluate exponents | $(15^{th}$ ↗$3^{rd})$ |
| Llemma 7B | 6.EE.A.1 - Evaluate exponents | $(18^{th}$ ↗$5^{th})$ |
| Gemma 2B | 8.EE.C.8 - Solve two-variable systems | $(22^{th}$ ↗$11^{th})$ |

Table 4: Model performance on our mathematical dialogue task, where the model must answer follow-up questions besides the initial problem. The second column, **Acc**uracy **with follow-ups**, shows overall success rate across standards that contain follow-up questions, considering a model successful only when it answers a problem and its follow-up questions correctly. The third and fourth columns show the hardest standard for each model when it comes to follow-up questions, showing a standard's code and abbreviated description, the model's accuracy ignoring follow-ups, and after follow-ups.

| Model | Acc. with follow-ups | Largest accuracy drop w/ follow-ups | |
|---|---|---|---|
| GPT-4o | 0.82 | 5.NF.A.1 - Add/sub fractions | 0.90 ↘0.61) |
| Claude-3 Opus | 0.76 | 7.NS.A.1-fraction - Add/sub with fractions | 0.57 ↘0.25) |
| Gemini-1.5 Pro | 0.77 | 5.NF.A.1 - Add/sub fractions | 0.60 ↘0.35) |
| Gemini-1.5 Flash | 0.76 | 7.NS.A.1-fraction - Add/sub with fractions | 0.78 ↘0.38) |
| GPT-3.5 Turbo | 0.71 | 7.NS.A.1-fraction - Add/sub with fractions | 0.73 ↘0.22) |
| Claude-3 Sonnet | 0.72 | 5.NF.A.1 - Add/sub fractions | 0.41 ↘0.07) |
| Claude-3 Haiku | 0.70 | 3.OA.A.3 - Mul/div within 100 | 1.00 ↘0.73) |
| Llama 3 70B | 0.69 | 4.NF.A.2 - Compare two fractions | 0.99 ↘0.66) |
| Mixtral 8x22B | 0.69 | 7.NS.A.1-fraction - Add/sub with fractions | 0.69 ↘0.18) |
| DeepSeek 67B | 0.68 | 6.NS.B.3 - Add/sub/mult/div decimals | 0.59 ↘0.37) |
| Llama 3 8B | 0.58 | 4.NF.A.2 - Compare two fractions | 0.90 ↘0.52) |
| Mixtral 8x7B | 0.58 | 5.NF.B.4 - Mult fractions | 0.61 ↘0.31) |
| Llemma 34B | 0.55 | 5.NF.B.4 - Mult fractions | 0.69 ↘0.33) |
| Mistral 7B | 0.48 | 7.NS.A.1-decimal - Add/sub with decimals | 0.91 ↘0.50) |
| DeepSeek Coder 33B | 0.60 | 3.OA.A.3 - Mul/div within 100 | 0.95 ↘0.81) |
| CodeLlama 34B | 0.60 | 5.NF.B.4 - Mult fractions | 0.52 ↘0.39) |
| phi-2 | 0.39 | 7.NS.A.2 - Mult/div with fractions | 0.57 ↘0.08) |
| Llemma 7B | 0.43 | 5.NF.B.4 - Mult fractions | 0.61 ↘0.22) |
| Gemma 7B | 0.33 | 7.NS.A.1-decimal - Add/sub with decimals | 0.91 ↘0.32) |
| CodeLlama 13B | 0.43 | 4.NBT.B.4 - Add/sub multi-digit nums | 0.81 ↘0.49) |
| CodeLlama 7B | 0.49 | 2.NBT.B.7 - Add/sub within 100 | 0.80 ↘0.67) |
| Gemma 2B | 0.24 | 3.NBT.A.2 - Add/sub within 1000 | 0.93 ↘0.26) |

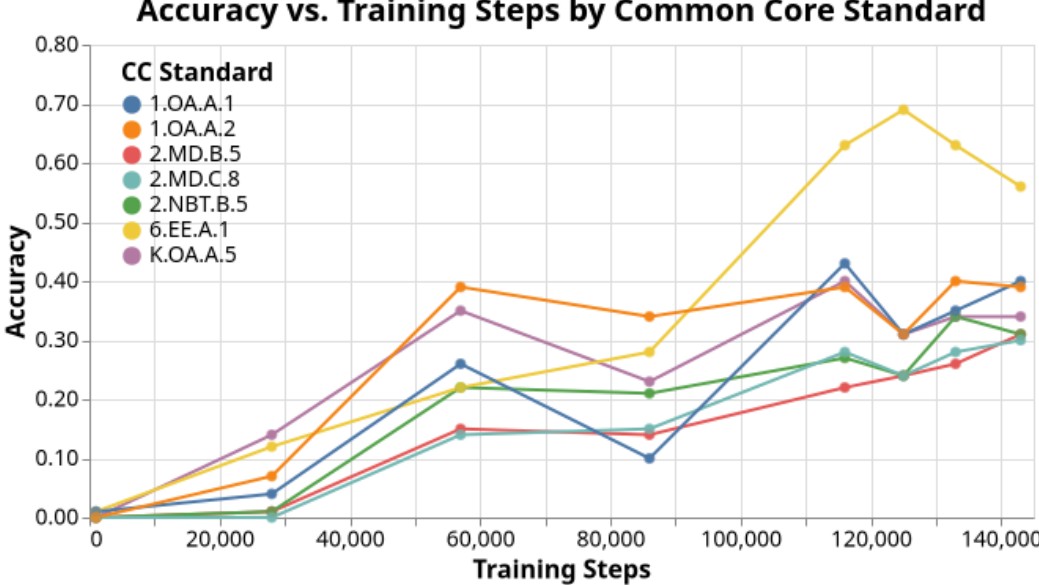

Figure 2: Performance of Pythia 12B checkpoints on MathCAMPS standards as it evolves during training. We show all 7 standards where the last checkpoint has at least 30% accuracy.

## B   Learning dynamics

We use Pythia [6] to showcase another analysis that MathCAMPS enables: understanding the learning dynamics of mathematical skills during LM training. We evaluate checkpoints of Pythia 12B on all standards, and track the performance change as the model was trained. Figure 2 shows Pythia's performance evolving during training on all 7 CC standards where the last checkpoint achieves at least 30% accuracy. Early in training, after 28k steps, Pythia performs best in a Kindergarten standard, K.OA.A.5 — "Fluently add and subtract within 5.". At 57k steps, its performance is best in both K.OA.A.5 (37% accuracy) and two first-grade standards, 1.OA.A.1 and 1.OA.A.2 — both standards involve simple word problems with addition and subtraction within 20. Pythia starts to become proficient at a sixth-grade standard around midway during training: 6.EE.A.1, which involves evaluating simple expressions using whole-number exponents (e.g, computing squares and cubes). These skills develop in tandem with its linguistic competence – at first, Pythia repeats questions verbatim often, but at 57k steps it already often produces *responses*. Overall, the high-resolution of MathCAMPS as a reasoning benchmark can support future work to deepen our understanding of how language models acquire capabilities during training, and how specific factors (such as data, or scale) contribute to their learning.

## C   Common Core Standards in MathCAMPS

MathCAMPS is available on Github at `https://github.com/gpoesia/mathcamps`. All of the Common Core standards we implement are described in a configuration file, `commoncore.yaml`, where standards are instantiated by composing high-level components from the Common Core attribute grammar. Moreover, we provide our prompts used to generate the dataset and model responses, as well as all problems and model responses for all LLMs we evaluated.

We list the Common Core standards we represent in MathCAMPS in Tables 5 through 13, segregated by grade. Standards 3.MD.D.8, 4.MD.A.2, 7.NS.A.1, and 7.NS.A.3 are split up into sub-standards. This was done for ease of implementation of the grammar.

## D   Familiarity bias

MathCAMPS was generated using GPT-4. GPT-4o, a model of the same family, was also the best performer overall (Table 1). To test whether this might be due to a familiarity bias — problems being

| Standard ID | Description |
|---|---|
| K.CC.C.7 | Compare two numbers between 1 and 10 presented as written numerals. |
| K.OA.A.4 | For any number from 1 to 9, find the number that makes 10 when added to the given number, e.g., by using objects or drawings, and record the answer with a drawing or equation. |
| K.OA.A.5 | Fluently add and subtract within 5. |
| K.NBT.A.1 | Compose and decompose numbers from 11 to 19 Into ten ones and some further ones, e.g., by using objects or drawings, and record each composition or decomposition by a drawing or equation (e.g., 18 = 10 + 8); understand that these numbers are composed of ten ones and one, two, three, four, five, six, seven, eight, or nine ones. |

Table 5: CC Standards for Grade K

| Standard ID | Description |
|---|---|
| 1.OA.A.1 | Use addition and subtraction within 20 to solve word problems involving situations of adding to, taking from, putting together, taking apart, and comparing, with unknowns in all positions, e.g., by using objects, drawings, and equations with a symbol for the unknown number to represent the problem. |
| 1.OA.A.2 | Solve word problems that call for addition of three whole numbers whose sum is less than or equal to 20, e.g., by using objects, drawings, and equations with a symbol for the unknown number to represent the problem. |
| 1.OA.D.8 | Determine the unknown whole number in an addition or subtraction equation relating three whole numbers. |

Table 6: CC Standards for Grade 1

| Standard ID | Description |
|---|---|
| 2.OA.A.1 | Use addition and subtraction within 100 to solve one- and two-step word problems involving situations of adding to, taking from, putting together, taking apart, and comparing, with unknowns in all positions, e.g., by using drawings and equations with a symbol for the unknown number to represent the problem. |
| 2.NBT.B.5 | Fluently add and subtract within 100 using strategies based on place value, properties of operations, and/or the relationship between addition and subtraction. |
| 2.NBT.B.6 | Add up to four two-digit numbers using strategies based on place value and properties of operations. |
| 2.NBT.B.7 | Add and subtract within 1000, using concrete models or drawings and strategies based on place value, properties of operations, and/or the relationship between addition and subtraction; relate the strategy to a written method. Understand that in adding or subtracting three-digit numbers, one adds or subtracts hundreds and hundreds, tens and tens, ones and ones; and sometimes it is necessary to compose or decompose tens or hundreds. |
| 2.MD.B.5 | Use addition and subtraction within 100 to solve word problems involving lengths that are given in the same units, e.g., by using drawings (such as drawings of rulers) and equations with a symbol for the unknown number to represent the problem. |
| 2.MD.C.8 | Solve word problems involving dollar bills, quarters, dimes, nickels, and pennies, using $ and ¢ symbols appropriately. |

Table 7: CC Standards for Grade 2

| Standard ID | Description |
|---|---|
| 3.OA.A.3 | Use multiplication and division within 100 to solve word problems in situations involving equal groups, arrays, and measurement quantities, e.g., by using drawings and equations with a symbol for the unknown number to represent the problem. |
| 3.OA.A.4 | Determine the unknown whole number in a multiplication or division equation relating three whole numbers. |
| 3.OA.C.7 | Fluently multiply and divide within 100, using strategies such as the relationship between multiplication and division (e.g., knowing that $8 \times 5 = 40$, one knows $40 \div 5 = 8$) or properties of operations. By the end of Grade 3, know from memory all products of two one-digit numbers. |
| 3.OA.D.8 | Solve two-step word problems using the four operations. Represent these problems using equations with a letter standing for the unknown quantity. Assess the reasonableness of answers using mental computation and estimation strategies including rounding. |
| 3.MD.D.8-triangle | Solve real world and mathematical problems involving perimeters of polygons, including finding the perimeter given the side lengths, finding an unknown side length, and exhibiting rectangles with the same perimeter and different areas or with the same area and different perimeters. |
| 3.MD.D.8-quadrilateral | Solve real world and mathematical problems involving perimeters of polygons, including finding the perimeter given the side lengths, finding an unknown side length, and exhibiting rectangles with the same perimeter and different areas or with the same area and different perimeters. |
| 3.MD.D.8-polygon | Solve real world and mathematical problems involving perimeters of polygons, including finding the perimeter given the side lengths, finding an unknown side length, and exhibiting rectangles with the same perimeter and different areas or with the same area and different perimeters. |
| 3.NBT.A.2 | Fluently add and subtract within 1000 using strategies and algorithms based on place value, properties of operations, and/or the relationship between addition and subtraction. |

Table 8: CC Standards for Grade 3

in-distribution for GPT-4o, but out-of-distribution for other models —, we generated a 10%-scale dataset using the exact same pipeline, but using Claude 3 Opus for both generating word problems and testing cycle consistency. This dataset has the same distribution of standards as MathCAMPS. We evaluated GPT-4o and Claude 3 Opus on this dataset — accuracies are reported in Table 14. GPT-4o also performs better in this dataset, suggesting that its performance in MathCAMPS was not due to a higher relative familiarity with the problems.

| Standard ID | Description |
|---|---|
| 4.OA.A.3 | Solve multistep word problems posed with whole numbers and having whole-number answers using the four operations, including problems in which remainders must be Interpreted. Represent these problems using equations with a letter standing for the unknown quantity. Assess the reasonableness of answers using mental computation and estimation strategies including rounding. |
| 4.OA.B.4 | Find all factor pairs for a whole number in the range 1-100. Recognize that a whole number is a multiple of each of its factors. Determine whether a given whole number in the range 1-100 is a multiple of a given one-digit number. Determine whether a given whole number in the range 1-100 is prime or composite. |
| 4.NBT.B.4 | Fluently add and subtract multi-digit whole numbers using the standard algorithm. |
| 4.NBT.B.5 | Multiply a whole number of up to four digits by a one-digit whole number, and multiply two two-digit numbers, using strategies based on place value and the properties of operations. Illustrate and explain the calculation by using equations, rectangular arrays, and/or area models. |
| 4.NBT.B.6 | Find whole-number quotients and remainders with up to four-digit dividends and one-digit divisors, using strategies based on place value, the properties of operations, and/or the relationship between multiplication and division. Illustrate and explain the calculation by using equations, rectangular arrays, and/or area models. |
| 4.NF.A.2 | Compare two fractions with different numerators and different denominators, e.g., by creating common denominators or numerators, or by comparing to a benchmark fraction such as 1/2. Recognize that comparisons are valid only when the two fractions refer to the same whole. Record the results of comparisons with symbols >, =, or <, and justify the conclusions, e.g., by using a visual fraction model. |
| 4.MD.A.2-decimal | Use the four operations to solve word problems involving distances, Intervals of time, liquid volumes, masses of objects, and money, including problems involving simple fractions or decimals, and problems that require expressing measurements given in a larger unit in terms of a smaller unit. Represent measurement quantities using diagrams such as number line diagrams that feature a measurement scale. |
| 4.MD.A.2-fraction | Use the four operations to solve word problems involving distances, Intervals of time, liquid volumes, masses of objects, and money, including problems involving simple fractions or decimals, and problems that require expressing measurements given in a larger unit in terms of a smaller unit. Represent measurement quantities using diagrams such as number line diagrams that feature a measurement scale. |
| 4.MD.A.3 | Apply the area and perimeter formulas for rectangles in real world and mathematical problems. |

Table 9: CC Standards for Grade 4

| Standard ID | Description |
| --- | --- |
| 5.OA.A.1 | Use parentheses, brackets, or braces in numerical expressions, and evaluate expressions with these symbols. |
| 5.NBT.B.5 | Fluently multiply multi-digit whole numbers using the standard algorithm. |
| 5.NBT.B.6 | Find whole-number quotients of whole numbers with up to four-digit dividends and two-digit divisors, using strategies based on place value, the properties of operations, and/or the relationship between multiplication and division. Illustrate and explain the calculation by using equations, rectangular arrays, and/or area models. |
| 5.NBT.B.7 | Add, subtract, multiply, and divide decimals to hundredths, using concrete models or drawings and strategies based on place value, properties of operations, and/or the relationship between addition and subtraction; relate the strategy to a written method and explain the reasoning used. |
| 5.NF.A.1 | Add and subtract fractions with unlike denominators (including mixed numbers) by replacing given fractions with equivalent fractions in such a way as to produce an equivalent sum or difference of fractions with like denominators. |
| 5.NF.A.2 | Solve word problems involving addition and subtraction of fractions referring to the same whole, including cases of unlike denominators, e.g., by using visual fraction models or equations to represent the problem. Use benchmark fractions and number sense of fractions to estimate mentally and assess the reasonableness of answers. |
| 5.NF.B.4 | Apply and extend previous understandings of multiplication to multiply a fraction or whole number by a fraction. |

Table 10: CC Standards for Grade 5

| Standard ID | Description |
| --- | --- |
| 6.NS.B.2 | Fluently divide multi-digit numbers using the standard algorithm. |
| 6.NS.B.3 | Add, subtract, multiply, and divide decimals to hundredths, using concrete models or drawings and strategies based on place value, properties of operations, and/or the relationship between addition and subtraction; relate the strategy to a written method and explain the reasoning used. |
| 6.EE.A.1 | Write and evaluate numerical expressions involving whole-number exponents. |
| 6.EE.B.7 | Solve real-world and mathematical problems by writing and solving equations of the form $x + p = q$ and $px = q$ for cases in which p, q and x are all nonnegative rational numbers. |

Table 11: CC Standards for Grade 6

| Standard ID | Description |
| --- | --- |
| 7.NS.A.1-fraction | Apply and extend previous understandings of addition and subtraction to add and subtract rational numbers; represent addition and subtraction on a horizontal or vertical number line diagram. |
| 7.NS.A.1-decimal | Apply and extend previous understandings of addition and subtraction to add and subtract rational numbers; represent addition and subtraction on a horizontal or vertical number line diagram. |
| 7.NS.A.2 | Apply and extend previous understandings of multiplication and division and of fractions to multiply and divide rational numbers. |
| 7.NS.A.3-fraction | Solve real-world and mathematical problems involving the four operations with rational numbers. |
| 7.NS.A.3-decimal | Solve real-world and mathematical problems involving the four operations with rational numbers. |

Table 12: CC Standards for Grade 7

| Standard ID | Description |
|---|---|
| 8.EE.A.2 | Use square root and cube root symbols to represent solutions to equations of the form $x^2 = p$ and $x^3 = p$, where p is a positive rational number. Evaluate square roots of small perfect squares and cube roots of small perfect cubes. Know that the square root of 2 is irrational. |
| 8.EE.C.7 | Solve linear equations in one variable. |
| 8.EE.C.8 | Analyze and solve pairs of simultaneous linear equations. |

Table 13: CC Standards for Grade 8

| Model | GPT4-generated MathCAMPS accuracy | Claude-generated MathCAMPS accuracy |
|---|---|---|
| GPT-4o | 0.910 | 0.954 |
| Claude 3 Opus | 0.887 | 0.909 |

Table 14: Performance of GPT-4o and Claude 3 Opus on the dataset genreated using Claude

# E  Data generation pipeline details

## E.1  Grammar

We implemented a global attribute grammar in Python, where production rules are implemented as recursive Python functions. Effectively, each CC standard has its own grammar, composed of pieces from components from the global CC grammar, as well as possibly adding unique non-terminals. Each CC standard contains the following parameters:

**Description:**  The description of the CC standard.

**Short description:**  A shortened description of the CC standard.

**Filters:**  A list of problem filters to ensure that all problems in this standard satisfy some requirement given in the Common Core description of the standard. The ProblemLength filter makes sure that the problem is within the desired length. CheckIntermediateValues filters out any problems with intermediate values greater or lesser than max_value or min_value, respectively. The ChainsOfVariables filter eliminates any problems where variables are assigned to equal exactly another variable, and nothing else. The ContainsTen filter checks if the math word problem contains numbers adding up to 10, or contains a 10 in the problem (for standards K.OA.A.4 and K.NBT.A.1, respectively).

**Transforms:**  List of problem transformations applied to all symbolic structures from this standard. The NoUselessVariables transform performs dead code elimination — it removes any variables that do not contribute to the final answer by applying a simple graph reachability algorithm on a dependency graph between statements, removing statements that the answer does not depend on. The Simplify transform essentially inlines variables that are used only once.

**Expressions:**  Lists non-terminals available to generate expressions in symbolic structures for this standard. For example, this can make specific binary operations (e.g. addition, division) available on that particular standard.

**Min/max value:**  Specifies bounds on values for both the final answer and all intermediate values in the solution.

**Min/max number:**  Specifies bounds on numeric constants sampled in the symbolic structure.

**Max depth:**  Sets a maximum depth for expressions in the symbolic structure.

**Samples:**  We include 2+ hand-written, standard-relevant examples of a symbolic problem followed by a relevant natural language problem generation, which we use as few-shot prompts during problem generation. We also use these prompts, but in reverse (natural language followed by symbolic problem), when we prompt GPT-4 during cycle consistency.

### E.2 Answer Grading During Evaluation

Given a solution in natural language, we first use a rule-based answer extractor to extract any model's numerical answer. In cases where a language model doesn't answer in the required format, or answers in an unexpected format, the answer is initially marked as incorrect. For all problems with incorrect answers, we use Llama-3 70B to re-extract the final answer. We few-shot prompt it with hand-generated examples of solutions and extracted final answers, and ask it to extract the final answer from the new solution. If a problem that was previously incorrect is marked as correct (given the newly extracted answer), we rerun the model on any followups the problem might have. Note that this "regrading" step can only improve accuracy from the base result, since we only run it on problems that failed under the rule-based evaluation. In practice, we found this process to have negligible false-positive rate — only in a handful of cases across all models we observed either answer extraction processes extracting the correct answer out of a wrong response (e.g., if the answer to a problem is 2, and the model responds "On day 2, Sally bought 9 dolls", the rule-based parser extracts 2 as being the model's answer, though the sentence implies its answer to be 9). On the other hand, the LLaMA-3 70B extractor greatly reduces our false negative rate in a handful of models (especially DeepSeek) which are more likely to respond in a format different from what our prompt asks for.

### E.3 Cost estimate

All problems in MathCAMPS were generated using OpenAI `gpt-4-0613`, in May 2024. We estimate an approximate cost of 330 USD to generate 9607 problems (including main problems and follow-ups). This includes the cost to perform cycle consistency, and problems that are discarded by cycle consistency. This gives an average cost of 0.034 USD (3.4 cents) per cycle-consistent problem or follow-up question.

## F Correlation between MathCAMPS and GSM8k

Figure 3 shows accuracies of several models on both GSM8k and MathCAMPS, along with the line of best fit. There is a strong correlation between overall accuracy in both datasets ($\rho = 0.91$, $p < 10^{-6}$), though MathCAMPS allows for many fine-grained analysis besides overall performance.

## G Largest Model Rank Changes When Focusing on One CC Standard (Complete Table)

Table 15 shows the full table from which Table 3 was extracted.

## H Followup Analysis

Table 16 lists model accuracies when only looking at the main problems (Main Acc.), their accuracies when only looking at the incremental followups (IFUP Acc.), their accuracies when only looking at the counterfactual followups (CFUP Acc.), and finally, the total number of followups seen by each model. The total number of followups a model sees relies on whether or not they get the main question for that followup correct. If a model does not correctly solve the main question, it is not prompted with follow-ups. Note that each followup serves as a followup to the main question, as opposed to a followup to each other.

| Model | Top outlier skill | Rank change |
|---|---|---|
| GPT-4o | 8.EE.C.8 - Solve two-variable systems | $(1^{st} \searrow 22^{th})$ |
| Claude-3 Opus | 2.MD.B.5 - Add/sub within 100 | $(2^{nd} \searrow 13^{th})$ |
| Gemini-1.5 Pro | K.OA.A.4 - Adding to equal 10 | $(3^{rd} \searrow 19^{th})$ |
| Gemini-1.5 Flash | 4.OA.B.4 - Factor pairs within 100 | $(4^{th} \searrow 20^{th})$ |
| GPT-3.5 Turbo | 6.EE.A.1 - Evaluate exponents | $(5^{th} \searrow 21^{th})$ |
| Claude-3 Sonnet | 2.NBT.B.5 - Add/sub within 100 | $(6^{th} \searrow 12^{th})$ |
| Claude-3 Haiku | 3.OA.A.4 - Determine unknowns in mul/div probs | $(9^{th} \nearrow 1^{st})$ |
| Llama 3 70B | K.OA.A.4 - Adding to equal 10 | $(7^{th} \searrow 17^{th})$ |
| Mixtral 8x22B | 8.EE.C.8 - Solve two-variable systems | $(8^{th} \searrow 21^{th})$ |
| DeepSeek 67B | K.NBT.A.1 - Decompose into 10s | $(10^{th} \nearrow 1^{st})$ |
| Llama 3 8B | 4.NBT.B.4 - Add/sub multi-digit nums | $(11^{th} \searrow 21^{th})$ |
| Mixtral 8x7B | 6.EE.A.1 - Evaluate exponents | $(12^{th} \searrow 20^{th})$ |
| Llemma 34B | K.OA.A.4 - Adding to equal 10 | $(13^{th} \nearrow 1^{st})$ |
| Mistral 7B | 1.OA.A.1 - Add/sub within 20 | $(14^{th} \searrow 21^{th})$ |
| DeepSeek Coder 33B | 6.EE.A.1 - Evaluate exponents | $(15^{th} \nearrow 3^{rd})$ |
| CodeLlama 34B | 5.NF.A.1 - Add/sub fractions | $(16^{th} \searrow 22^{th})$ |
| phi-2 | K.OA.A.4 - Adding to equal 10 | $(17^{th} \nearrow 4^{th})$ |
| Llemma 7B | 6.EE.A.1 - Evaluate exponents | $(18^{th} \nearrow 5^{th})$ |
| Gemma 7B | K.OA.A.5 - Add/sub within 5 | $(19^{th} \nearrow 6^{th})$ |
| CodeLlama 7B | 8.EE.C.8 - Solve two-variable systems | $(21^{th} \nearrow 15^{th})$ |
| Gemma 2B | 8.EE.C.8 - Solve two-variable systems | $(22^{th} \nearrow 11^{th})$ |

Table 15: Largest changes in a model's ranking when comparing its performance in a particular CC standard, in contrast to only overall performance. This is a complete version of Table 3, which only showed some models for brevity.

| Vendor | Model | Main Acc. | IFUP Acc. | CFUP Acc. | Total FUPs seen |
|---|---|---|---|---|---|
| Anthropic | Claude-3 Opus | 0.89 | 0.90 | 0.88 | 4142 |
| Anthropic | Claude-3 Sonnet | 0.86 | 0.86 | 0.87 | 3964 |
| Anthropic | Claude-3 Haiku | 0.84 | 0.88 | 0.87 | 3819 |
| DeepSeek | DeepSeek Coder 33B | 0.65 | 0.79 | 0.85 | 1022 |
| DeepSeek | DeepSeek 67B | 0.80 | 0.87 | 0.88 | 3286 |
| EleutherAI | LLemma 7B | 0.62 | 0.68 | 0.80 | 2890 |
| Google | Gemini-1.5 Pro | 0.89 | 0.91 | 0.89 | 4140 |
| Google | Gemini-1.5 Flash | 0.87 | 0.89 | 0.87 | 4083 |
| Google | Gemma 2B | 0.51 | 0.29 | 0.54 | 2044 |
| Google | Gemma 7B | 0.62 | 0.55 | 0.60 | 2786 |
| Meta | Llama 3 8B | 0.77 | 0.84 | 0.80 | 3476 |
| Meta | Llama 3 70B | 0.85 | 0.87 | 0.84 | 3939 |
| Meta | CodeLlama 7B | 0.52 | 0.69 | 0.86 | 617 |
| Meta | CodeLlama 13B | 0.58 | 0.75 | 0.80 | 2451 |
| Meta | CodeLlama 34B | 0.64 | 0.82 | 0.88 | 844 |
| Microsoft | phi-2 | 0.63 | 0.48 | 0.78 | 2873 |
| Mistral | Mistral 7B | 0.68 | 0.72 | 0.80 | 3090 |
| Mistral | Mixtral 8x7B | 0.76 | 0.80 | 0.82 | 3439 |
| Mistral | Mixtral 8x22B | 0.84 | 0.86 | 0.83 | 3948 |
| OpenAI | GPT-4o | 0.92 | 0.92 | 0.90 | 4358 |
| OpenAI | GPT-3.5 Turbo | 0.87 | 0.85 | 0.86 | 4063 |

Table 16: Accuracy of each model on *incremental* follow-up questions (**IFUP**) as well as on *counterfactual* follow-ups (**CFUP**). Note that these accuracies are not directly comparable, since models are only evaluated on follow-ups to problems that they respond correctly to; thus, each accuracy shown here is over a different subset of follow-up problems in MathCAMPS.

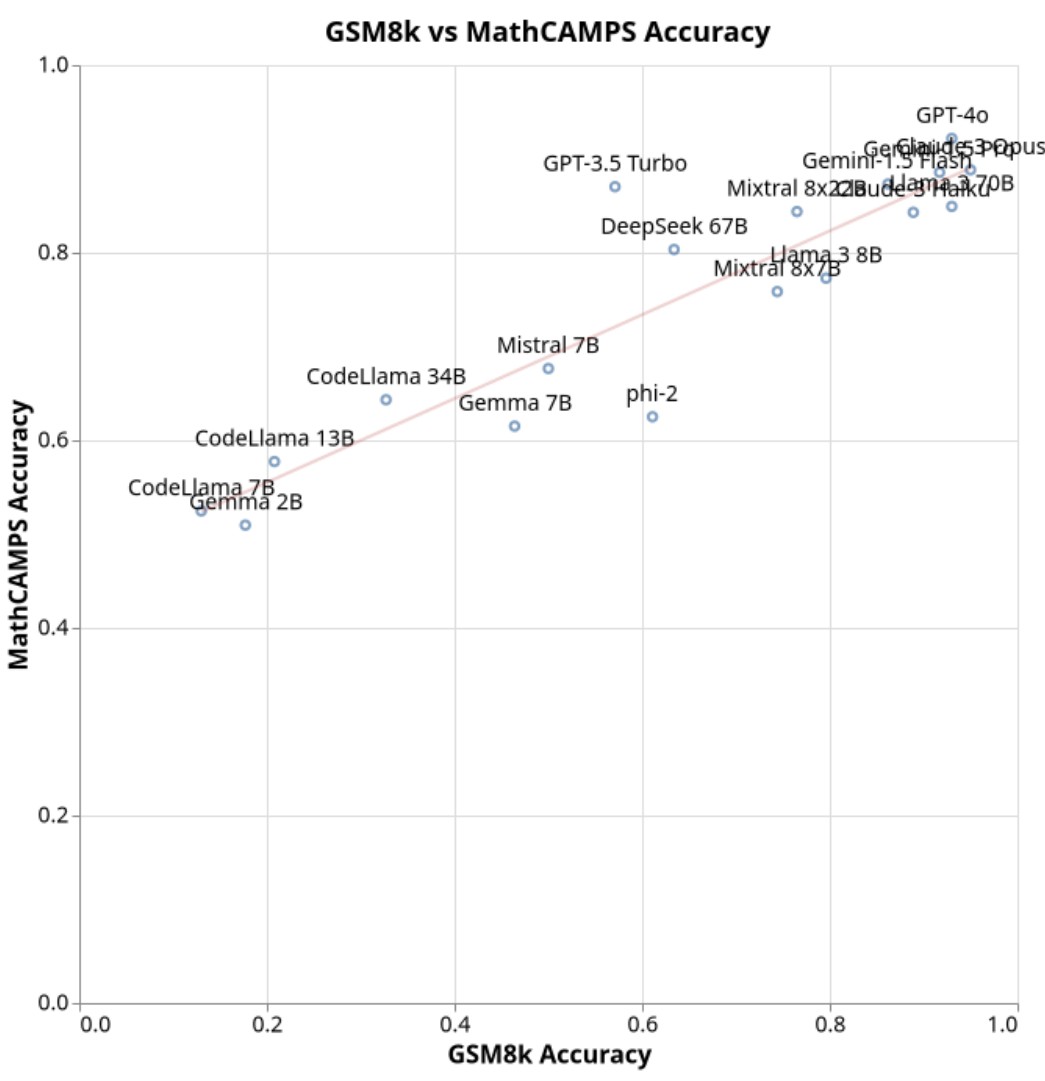

Figure 3: Relation between accuracy on GSM8k and on MathCAMPS.

