# OpenReview forum: "MathCAMPS: Fine-grained Synthesis of Mathematical Problems From Human Curricula"
_NeurIPS.cc/2024/Workshop/MATH-AI — MATH-AI 24_

### Official Review · Reviewer_LDi3 · 2024-10-04
**The proposed dataset has questions which data quality and difficulty are not up-to today's standards**

**Rating:** 3
**Confidence:** 5

**Review:**

The paper proposes a new dataset to evaluate grade-shool math skills of LLMs with fine-grained categories. While breaking down evaluation by categories is certainly useful, I think that focusing on grade-school math is not relevant anymore as current frontier LLMs (both closed and open-weights) have almost perfect accuracy on those questions (e.g. judging by gsm8k benchmark).

That's why I was surprised to see in Table 1 of the paper that the highest score for K=8 was just 64%. The paper has a link to github (is that allowed under the workshop rules?) which I used to look at the dataset questions. I reviewed 10 samples where gpt-4o made mistakes and found that 8/10 questions were incorrect, ambiguous or the LLM answer was correct, but not extracted properly. This makes me question the dataset quality (especially for higher grades) and thus any conclusions we might draw from the presented evaluation.

I think that the authors have a good idea to create new math benchmark with carefully labeled categories, but the presented work fails in both quality and the difficulty levels of the provided questions. I would encourage authors to instead focus on harder kinds of problems (e.g. MATH level or even higher) and validate the quality of their dataset more rigorously.

---

### Official Review · Reviewer_5egw · 2024-10-07
**This paper introduce a framework for generating the high-quality problems and evaluated on 23 models.**

**Rating:** 6
**Confidence:** 3

**Review:**

Questions: Do you think the gap between open-source models and closed-source models is due to differences in long-context following or problem-solving capabilities?

Pros:
1. The authors propose a framework for generating high-quality mathematical problems based on Common Core (CC) standards.
2. The evaluation on 23 LLMs provides comprehensive results.

Cons:
1. The connection between follow-up questions and the human curriculum is somewhat unclear.
2 The main contribution of this paper is not well highlighted. Many contributions seem equally emphasized.

If you can address all my concerns, I'm happy to change my score.

---

### Official Review · Reviewer_nDsv · 2024-10-08
**Review of "MathCAMPS: Fine-grained Synthesis of Mathematical Problems From Human Curricula"**

**Rating:** 8
**Confidence:** 4

**Review:**

# Summary
The paper "MathCAMPS: Fine-grained Synthesis of Mathematical Problems From Human Curricula" addresses a critical issue in the training of Large Language Models (LLMs) — the potential data contamination due to the models being trained on vast public datasets that may include benchmark datasets. The authors propose two main contributions to tackle this issue:
1. Creation of a Large Fine-grained Synthetic Benchmark: The paper introduces a synthetic benchmark specifically designed for evaluating LLMs on mathematical reasoning. This benchmark is based on K-8 school Common Core math questions, providing a fine-grained and relevant dataset for testing.
2. Methodology Using Follow-up Questions: To enhance the evaluation of LLMs, the authors propose a novel methodology that incorporates follow-up questions to increase cycle-consistency. This approach helps in assessing the model's ability to maintain logical consistency over sequences of related questions.

The study finds that most LLM models exhibit lower performance when evaluated using this new methodology, highlighting the challenges in current LLM capabilities concerning mathematical reasoning.

# Evaluation
The paper is exceptionally well-written, with clear explanations that make the concepts accessible to anyone with a reasonable mathematical background. The problem statement is well-articulated, and the proposed solutions are both innovative and directly address the identified issues.
The introduction of a fine-grained synthetic benchmark is particularly noteworthy as it fills a significant gap in the resources available for evaluating the mathematical reasoning capabilities of LLMs. This benchmark could serve as a valuable tool for future research and development in the field.
The methodology using follow-up questions is another significant contribution, offering a novel way to enhance the evaluation process of LLMs. This approach not only tests the models' immediate problem-solving capabilities but also their ability to engage in a form of 'sustained' reasoning, maintaining logical consistency across multiple interrelated problems.

# Recommendation
Given the clarity of the presentation, the relevance of the problem addressed, and the novelty of the contributions, I highly recommend the acceptance of this paper. The findings and resources presented could have substantial impacts on the field, particularly in improving how LLMs are trained and evaluated concerning mathematical reasoning. This work not only advances our understanding of LLM capabilities but also provides practical tools and methodologies that can be used to further refine these models.

---

### Official Review · Reviewer_ozAs · 2024-10-09
**This paper provides a good fine-grained synthesis benchmark for mathematical reasoning in LLMs.**

**Rating:** 6
**Confidence:** 3

**Review:**

Thank you for submitting this interesting paper. This work proposes a fine-grained synthesis benchmark for mathematical reasoning in LLMs, called MathCAMPs.

The cycle-consistency procedure introduced by the authors is encouraging, and maybe, expert feedbacks can also be taken into consideration for future iterations of this benchmark.

There is a somewhat abnormal point that is not explained in this paper. Why do the LLMs perform worse on grade-5 problems compared to grade-6 problems, while the latter should be more difficult? A discussion of the potential reasons will be a nice addition. (e.g., some topics included in grade-5 CC are special.)

Overall, I could see some potential in this paper to become a good benchmark in related work.

---

### Decision · Program_Chairs · 2024-10-09

Accept